# Predicting the Risk of Ischemic Stroke among Patients with Rheumatoid Arthritis Using a Simplified RA-CHADSV Score Based on the CHA_2_DS_2_-VASc Score

**DOI:** 10.3390/medicina56020073

**Published:** 2020-02-12

**Authors:** Chia-Wen Hsu, Khai-Jing Ng, Ming-Chi Lu, Malcolm Koo

**Affiliations:** 1Department of Medical Research, Dalin Tzu Chi Hospital, Buddhist Tzu Chi Medical Foundation, Dalin, 62247 Chiayi, Taiwan; chiawen0114@yahoo.com.tw; 2School of Nursing, College of Medicine, Chang Gung University, 33302 Taoyuan City, Taiwan; 3Division of Allergy, Immunology and Rheumatology, Dalin Tzu Chi Hospital, Buddhist Tzu Chi Medical Foundation, Dalin, 62247 Chiayi, Taiwan; khaijingcool88@gmail.com (K.-J.N.); e360187@yahoo.com.tw (M.-C.L.); 4School of Medicine, Tzu Chi University, Hualien City, 97004 Hualien, Taiwan; 5Graduate Institute of Long-term Care, Tzu Chi University of Science and Technology, Hualien City, 97005 Hualien, Taiwan; 6Dalla Lana School of Public Health, University of Toronto, Toronto, M5T 3M7 ON, Canada

**Keywords:** stroke, rheumatoid arthritis, atrial fibrillation, brain ischemia, risk assessment

## Abstract

Background and Objectives: The aim of this retrospective cohort study was to develop a new score (RA-CHADSV) (rheumatoid arthritis - congestive heart failure, hypertension, age ≥75 years, diabetes mellitus, stroke/transient ischemic attack/thromboembolism, and vascular disease), modified from the CHA_2_DS_2_-VASc score (congestive heart failure, hypertension, age ≥75 years (doubled), diabetes mellitus, stroke/transient ischemic attack (doubled), vascular disease, age 65–74 years, and female), in predicting the risk of ischemic stroke in rheumatoid arthritis (RA) patients without atrial fibrillation (AF). Materials and Methods: Using the Taiwan’s National Health Insurance Research Database, 592 patients with RA diagnosed between 2000 and 2002 were identified and followed until first occurrence of ischemic stroke or the last available date in the database. Incidence rate ratios (IRR) of ischemic stroke for the CHA_2_DS_2_-VASc score were calculated using Poisson regression models. A new prediction score RA-CHADSV was developed using multiple logistic regression analysis with bootstrap validation. Results: The area under the receiver operating characteristic curve of the newly developed RA-CHADSV score and the CHA_2_DS_2_-VASc score were 0.73 (95% confidence interval (CI) 0.64–0.82) and 0.70 (95% CI 0.61–0.79), respectively. The RA-CHADSV score was significantly associated with a higher ischemic stroke incidence in the patients who scored ≥1 (adjusted IRR 7.39, *p* < 0.001). Conclusions: A simplified RA-CHADSV score, with comparable efficiency as the CHA_2_DS_2_-VASc score, but easier to use clinically was developed for predicting the risk of ischemic stroke among non-AF RA patients.

## 1. Introduction

Rheumatoid arthritis (RA) is one of the most prevalent chronic inflammatory polyarthritis with a prevalence of 0.5–1% with female predominance in industrialized countries [1]. The risk of stroke is higher in patients with RA compared with the general population or patients with noninflammatory arthropathies [2,3,4,5]. Patients with RA are also at a higher risk for long-term disability after stroke [6]. Previous research has shown that the presence of autoantibodies, autoantigens, and pro-inflammatory cytokines in autoimmune diseases could accelerate the process of atherosclerosis [7,8]. Abnormal endothelial function and arterial intima-media thickening could cause the formation of atherosclerotic lesions [7], and these lesions are prone to rupture in patients with RA [8,9]. Therefore, patients with RA were associated with a higher risk of stroke [10,11], which might lead to long-term disability after stroke [6] and increased mortality [12].

Several numerical tools have recently been developed for the prediction of ischemic stroke [13,14]. The CHADS_2_ score (congestive heart failure, hypertension, age ≥75 years, diabetes mellitus, and stroke [doubled]) and its updated version CHA_2_DS_2_-VASc score (congestive heart failure, hypertension, age ≥75 years (doubled), diabetes mellitus, stroke/transient ischemic attack (doubled), vascular disease, age 65–74 years, and female) are commonly used clinical prediction rules for estimating the risk of major adverse cardiovascular events (MACE) among patients with atrial fibrillation (AF) [15,16,17,18]. These scores are also widely used in the decision making of atrial fibrillation ablation and non-valvular atrial fibrillation (NVAF) to predict the risk of MACE and stroke [19,20]. Recently, the CHA_2_DS_2_-VASc has been used for the evaluation of the risk of stroke and MACE in specific populations without AF [21,22], including heart failure, systemic lupus erythematosus, diabetes, hemodialysis, and RA [23,24,25,26]. We proposed that the risk of ischemic stroke might be different in patients with RA. The aim of this secondary retrospective cohort study was to develop a new and simplified score, based on the CHADS_2_ and CHA_2_DS_2_-VASc scores, for predicting the risk of ischemic stroke in RA patients without AF.

## 2. Materials and Methods

### 2.1. Data Source

This retrospective cohort study used data from the Longitudinal Health Insurance Database 2000 (LHID 2000). The LHID 2000 contains one million randomly selected patients, excluding medical personnel, from the year 2000 Registry of Beneficiaries of the Taiwan’s National Health Insurance Research Database. No statistical significant differences existed in the age, sex, or health-care costs between the LHID 2000 and all enrollees of the Taiwan’s National Health Insurance (NHI) program. The NHI program is a single-payer universal health insurance plan covering the health care services for over 99% of Taiwan’s residents [27]. The study protocol was approved by the Institutional Review Board of Dalin Tzu Chi Hospital, Buddhist Tzu Chi Medical Foundation, Taiwan (No. B10004021) on 20 January 2012.

### 2.2. Study Population

In this study, patients with RA were identified from the LHID 2000 using the International Classification of Disease, 9th Revision, Clinical Modification (ICD-9-CM) code 714.0, from 1 January 2000 to 31 December 2002. We excluded patients who were diagnosed with AF or atrial flutter (ICD-9-CM codes 427.31 and 427.32 [25]) and under 20 years of age. 

A total of 592 non-AF RA patients were included in the study, and 477 patients were females (80.6%), 460 patients were ≤64 year-old (77.7%), and 33.6% had at least one comorbidities. The two most common comorbidities were hypertension (20.8%) and diabetes mellitus (8.3%). The mean follow-up period was 9.97 years (Table 1). 

### 2.3. Outcome Measurement and Comorbidities

Patients were followed until first occurrence of ischemic stroke (ICD-9-CM codes 433.x1, 434 (excluding 434.x0), 435, and 436), [28,29] or the last available date in the database. The longest follow-up duration was up to 10 years. The CHA_2_DS_2_-VASc score was applied to evaluate the risk of ischemic stroke among RA patients without AF. 

The CHADS_2_ score was calculated according to the following algorithm with five variables: congestive heart failure (ICD-9-CM code 428, 1 point), hypertension (ICD-9-CM codes 401–405, 1 point), age ≥ 75 years (1 point), diabetes (ICD-9-CM code 250, 1 point), and prior ischemic stroke/transient ischemic attack (ICD-9-CM codes 433–438, 2 points). The CHA_2_DS_2_-VASc score was calculated according to the following algorithm with 8 variables: Congestive heart failure (ICD-9-CM code 428, 1 point), hypertension (ICD-9-CM codes 401–405, 1 point), diabetes (ICD-9-CM code 250, 1 point), age ≥ 75 years (2 points), prior ischemic stroke/transient ischemic attack (ICD-9-CM codes 433–438, 2 points), vascular disease (ICD-9-CM codes 410.x, 411.0, 412, 420.20–420.24, 429.79, 440.0, 440.20–440.24, 440.29, 440.30–440.32, 440.4, 443.81, 443.89, and 443.9), age 65–75 years (1 point), and female sex (1 point) [24,30,31].

In AF patients, a CHADS_2_ score ≥2 or a CHA_2_DS_2_-VASc score ≥2 was considered as high risk for stroke, respectively [32]. Comorbidities related to ischemic stroke, including hyperlipidemia (ICD-9-CM code 272), chronic obstructive pulmonary disease (ICD-9-CM codes 491, 492, and 496), and chronic kidney disease (ICD-9-CM codes 585, 586, 588.8, and 588.9), and obesity (ICD-9-CM code 278) were included as potential confounding variables [24,30,31,33,34,35,36].

### 2.4. Statistical Analysis

Baseline characteristics of the study participants are shown as means and standard deviations (SDs) or frequency and percentages for continuous and categorical variables, respectively. Receiver operating characteristic (ROC) curve was constructed to assess the predictive ability of the CHADS_2_ score and CHA_2_DS_2_-VASc score for ischemic stroke. The incidence rates per 10,000 person-years were calculated for ischemic stroke separately for the CHA_2_DS_2_-VASc score <2 cohort and the CHA_2_DS_2_-VASc score ≥2 cohort. The cumulative incidence of ischemic stroke in the non-AF RA patients for the two cohorts was determined using the Kaplan–Meier method and compared with log-rank test. The incidence rates and incidence rate ratios (IRRs) of the outcome variables, stratified by follow-up periods, between the CHA_2_DS_2_-VASc score 0–1 and ≥2 cohorts were calculated using univariate and multiple Poisson regression models (i.e., generalized linear outcome with a Poisson log-liner link function and person-years as the offset variable), adjusting for dyslipidemia, chronic obstructive pulmonary disease, chronic kidney disease, and obesity. 

Regarding the model development for the new RA-CHADSV score (congestive heart failure, hypertension, age ≥75 years, diabetes mellitus, stroke/transient ischemic attack/thromboembolism, and vascular disease), we used multiple logistic regression analysis to obtain regression coefficients. The model was validated internally with a bootstrap approach using 1000 samples by resampling with replacement from the original sample with bias-corrected and accelerated confidence interval [37]. The new model of RA-CHADSV score was calculated as the following algorithm: congestive heart failure (1 point), hypertension (1 point), age ≥ 75 years (1 point), diabetes mellitus (1 point), stroke/transient ischemic attack/thromboembolism (1 point), and vascular disease (1 point). All statistical analyses were performed using IBM SPSS Statistics for Windows, Version 24.0 (IBM Corp, Armonk, NY, USA). A two-tailed *p* value of <0.05 was considered statistically significant.

## 3. Results

For the CHA_2_DS_2_-VASc score, 396 patients score 0–1, and 196 patients score 2–9. The area under the curve (AUC) of the ROC curve of CHADS_2_ score was 0.52 (95% CI: 0.42–0.62, *p* = 0.671) (Figure A1) and that of the CHA_2_DS_2_-VASc score was 0.70 (95% CI: 0.61–0.79, *p* < 0.001) (Figure A2), which showed the ability of the two scores to discriminate ischemic stroke in non-AF RA patients. Figure 1 signifies that non-AF RA patients with CHA_2_DS_2_-VASc score ≥2 show significantly higher incidence rates of ischemic stroke compared to those with CHA_2_DS_2_-VASc score <2 (*p* < 0.001). 

Table 2 shows the incidence rates and IRRs of ischemic stroke for patients with CHA_2_DS_2_-VASc score 0–1 and ≥2. The incidence rates of ischemic stroke in non-AF RA patients with CHA_2_DS_2_-VASc score 0–1 and ≥2 were 23.81 and 146.89 per 10,000 person-years, respectively. A significantly higher incidence of ischemic stroke in patients who scored ≥2 was observed (adjusted IRR 6.00, *p* < 0.001). Table 2 also shows the incidence rates and IRRs of ischemic stroke with stratification by follow-up years. Significantly higher incidences of ischemic stroke were observed in patients with a CHA_2_DS_2_-VASc score ≥2 compared to those with a CHA_2_DS_2_-VASc score of 0 or 1. The adjusted IRR was 5.38 (*p* < 0.001) and 5.91 (*p* < 0.001) for a follow-up of 5 and 10 years, respectively.

A new and simplified score for improving the prediction of ischemic stroke risk in patients with non-AF RA based on the CHA_2_DS_2_-VASc score was developed and shown in Table 3. Based on the results of the logistic model obtained while simultaneously considering clinical convenience, variables with a logistic regression coefficient near zero were omitted from the final model (age 65–74 years and female). In addition, the regression coefficients were round to one significant figure for convenience. For congestive heart failure, we compared the AUC obtained based on a regression coefficient of 1 and 2, and found no differences between the two. Therefore, the variable congestive heart failure was given a score of 1 in the final model. Correspondingly, our new proposed score RA-CHADSV is named based on the acronym of the items in it: congestive heart failure (1 point), hypertension (1 point), age ≥ 75 years (1 point), diabetes mellitus (1 point), stroke/transient ischemic attack/thromboembolism (1 point), and vascular disease (1 point). A RA-CHADSV score ≥1 suggested a higher risk of ischemic stroke [26].

The AUC of ROC curve of the new RA-CHADSV score was 0.73 (95% CI: 0.64–0.82, *p* < 0.001) (Figure A3). Patients with RA-CHADSV score ≥1 showed a significantly higher incidence rate of ischemic stroke compared to those who scored 0 (*p* < 0.001) (Figure 2). Table 4 showed that with every one unit of increment in the RA-CHADSV score, the risk of ischemic stroke increased 2.49 times (*p* < 0.001). In addition, a significantly higher ischemic stroke incidence in the patients who scored ≥ 1, compared with those who scored 0, was observed with an adjusted IRR of 7.39 (*p* < 0.001). 

RA-CHADSV score was calculated according to: congestive heart failure (1 point), hypertension (1 point), age ≥ 75 (1 point), diabetes mellitus (1 point), stroke/transient ischemic attack/thromboembolism (1 point), and vascular disease (1 point).

## 4. Discussion

This retrospective cohort study has three main findings. First, the CHA_2_DS_2_-VASc score was able to moderately predict the risk of ischemic stroke in non-AF RA patients, with a score of ≥2 possessed a higher risk of ischemic stroke than patients with a score of 0–1. Second, the CHADS_2_ score inadequately predicted the risk of ischemic stroke in non-AF RA patients. Third, our newly developed RA-CHADSV score (consisted of six variables with equal weights of one: Congestive heart failure, hypertension, age ≥ 75 years, diabetes mellitus, stroke/transient ischemic attack/thromboembolism, and vascular disease) showed similar performance compared with the original CHA_2_DS_2_-VASc score in predicting the risk of ischemic stroke in non-AF RA patients. Our study results support the clinical utility of CHA_2_DS_2_-VASc score in prediction of the MACE in non-AF RA patients [26]. In addition, we showed that the new RA-CHADSV score was able to predict the risk of ischemic stroke in patients with non-AF RA.

Our results showed that non-AF RA patients possessed a significantly higher risk of ischemic stroke with a follow-up of ten years. The most common comorbidity in our patients was hypertension, which accounted for 33.6% of patients. This result was similar to a previous study where RA patients tended to develop more than one comorbidity within 5 years after diagnosis of RA, with the most common comorbidity was found to be hypertension [38]. 

The CHA_2_DS_2_-VASc score have been used extensively to predict the risk of ischemic stroke in certain individuals [17,20,23,25]. Previous research revealed that patients with RA were at a higher risk for cardiovascular disease compared to patients with other immune-mediated diseases [39,40]. Our finding that non-AF RA patients with a CHA_2_DS_2_-VASc score of ≥2 had a higher risk of ischemic stroke is similar to the case of systemic lupus erythematosus patients without AF [25]. Previous studies have also demonstrated an increased risk of ischemic stroke 10 or more years after RA diagnosis [3,4,41]. This result was also observed in our study, especially in patients with a higher CHA_2_DS_2_-VASc score (≥2). However, not all the parameters in the CHA_2_DS_2_-VASc score can be applicable for patients with RA, hence we developed a new score for non-AF RA patients based on multiple logistic regression analysis with bootstrapping internal validation. The ROC curve of RA-CHADSV and the CHA_2_DS_2_-VASc score was 0.73 and 0.70, respectively. Despite the simplicity (only six variables of equal weights) of our new model, the performance was comparable with the original CHA_2_DS_2_-VASc score in predicting ischemic stroke risk among non-AF RA patients. 

This study has a few limitations that should be noted. Patient’s physical activity, lifestyle factors, and laboratory data were not available in the LHID, and therefore could not be adjusted in the statistical models. The model for the new RA-CHADV score was validated internally using bootstrap method, which should be externally validated in future studies.

## 5. Conclusions

Findings from this retrospective cohort study showed that CHA_2_DS_2_-VASc score was able to moderately predict the risk of ischemic stroke in non-AF RA patients. Furthermore, a new RA-CHADSV score was developed in this study, which was based on six variables, including congestive heart failure, hypertension, age ≥ 75 years, diabetes mellitus, stroke/transient ischemic attack/thromboembolism, and vascular disease. This simple tool could be used to identify non-AF RA patients with an increased risk of ischemic stroke, and to help physicians in the management of these patients.

## Figures and Tables

**Figure 1 medicina-56-00073-f001:**
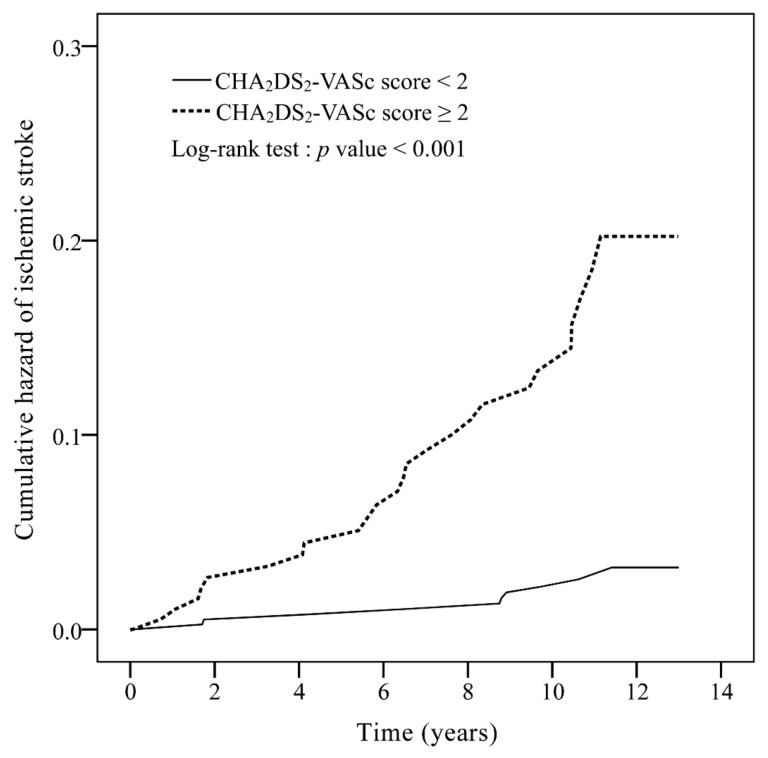
Cumulative incidence curves of ischemic stroke for rheumatoid arthritis patients without atrial fibrillation, with a CHA_2_DS_2_-VASc score 0–1 and CHA_2_DS_2_-VASc score ≥2. CHA_2_DS_2_-VASc: congestive heart failure, hypertension, age ≥75 years (doubled), diabetes mellitus, prior stroke or transient ischemic attack (doubled), vascular disease, age 65–74 years, and female.

**Figure 2 medicina-56-00073-f002:**
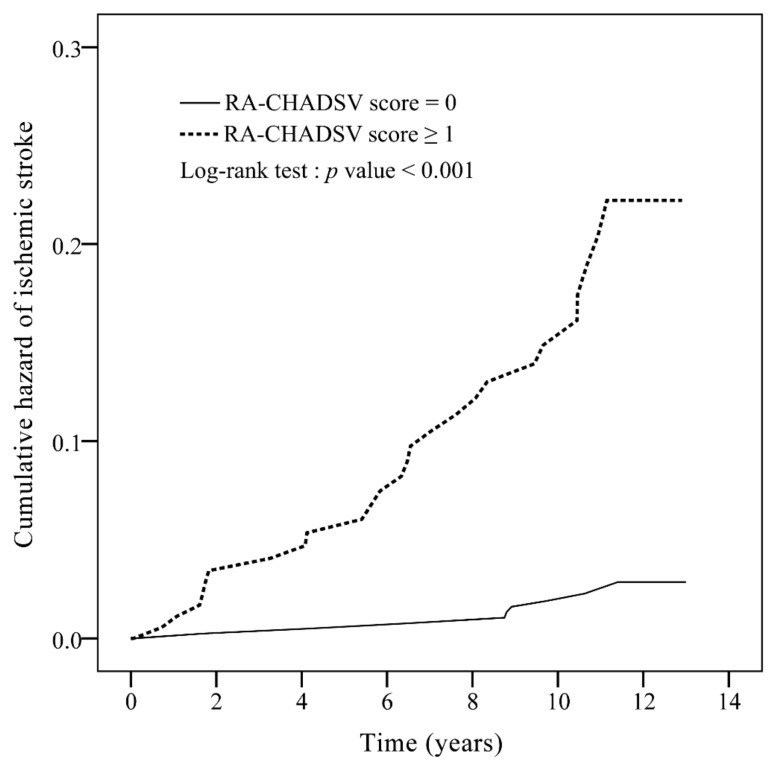
Cumulative incidence curves of ischemic stroke for rheumatoid arthritis patients without atrial fibrillation, with a RA-CHADSV score 0 and RA-CHADSV score ≥1. RA-CHADSV: rheumatoid arthritis - congestive heart failure, hypertension, age ≥75 years, diabetes mellitus, stroke/transient ischemic attack/thromboembolism, and vascular disease.

**Table 1 medicina-56-00073-t001:** Baseline characteristics of the study participants (N = 592).

Variable	*n*	(%)
Age, years		
≤64	460	(77.7)
65–74	87	(14.7)
≥75	45	(7.6)
Sex		
Female	477	(80.6)
Male	115	(19.4)
Comorbidities (at least one condition)	199	(33.6)
Hypertension	123	(20.8)
Diabetes mellitus	49	(8.3)
Congestive heart failure	2	(0.3)
Previous stroke/transient ischemic attack	7	(1.2)
Vascular disease	6	(1.0)
Hyperlipidemia	27	(4.6)
Chronic obstructive pulmonary disease	36	(6.1)
Chronic kidney disease	6	(1.0)
Obesity	1	(0.2)
Mean follow up, years (standard deviation)	9.97	(2.84)
CHA_2_DS_2_-VASc score		
Mean (standard deviation)	1.44	(1.05)
0–1	396	(66.9)
≥2	196	(33.1)

CHA_2_DS_2_-VASc: congestive heart failure, hypertension, age ≥75 years (doubled), diabetes mellitus, prior stroke or transient ischemic attack (doubled), vascular disease, age 65–74 years, and female.

**Table 2 medicina-56-00073-t002:** Incidence rates and incidence rate ratio of ischemic stroke for patients with rheumatoid arthritis according to CHA_2_DS_2_-VASc score ≥2, with stratification by follow up years (N = 592).

CHA_2_DS_2_-VASc Score	Follow Up	Events	Person-Years	IR (per 10,000 Person-Years)	IRR (95% CI) *p* Value	Adjusted IRR * (95% CI) *p* Value
Continuous score	Overall	35	5902	59.30	1.90 (1.52–2.37) <0.001	1.75 (1.39–2.19) <0.001
Dichotomized score						
0–1	Overall	10	4200	23.81	1	1
≥2	Overall	25	1702	146.89	6.10 (2.93–12.69) <0.001	6.00 (2.88–12.52) <0.001
Continuous score	5-year	11	2837	38.77	1.68 (1.11–2.54) 0.013	1.48 (0.98–2.23) 0.062
Dichotomized score						
0–1	5-year	3	1940	15.46	1	1
≥2	5-year	8	897	89.19	5.77 (1.53–21.75) 0.010	5.38 (1.42–20.36) 0.013
Continuous score	10-year	28	5306	52.77	1.87 (1.46–2.39) <0.001	1.70 (1.32–2.19) <0.001
Dichotomized score						
0–1	10-year	8	3738	21.40	1	1
≥2	10-year	20	1568	127.55	5.96 (2.62–13.53) <0.001	5.91 (2.60–13.45) <0.001

CI: confidence interval; IR: incidence rate; IRR: incidence rate ratio; CHA_2_DS_2_-VASc: congestive heart failure, hypertension, age ≥75 years (doubled), diabetes mellitus, stroke/transient ischemic attack (doubled), vascular disease, age 65–74 years, and female. * Adjusted for the patient dyslipidemia, chronic obstructive pulmonary disease, chronic kidney disease, and obesity.

**Table 3 medicina-56-00073-t003:** Multiple logistic regression model for the development of the retrospective cohort study was to develop a new score (RA-CHADSV) score.

Variable	Adjusted β	Standard Error	95% CI
Congestive heart failure	1.60	17.94	−19.12–42.26
Hypertension	1.32	0.41	0.49–2.23
Age ≥ 75 years	0.57	1.17	−1.04–1.70
Diabetes mellitus	1.31	0.52	0.06–2.45
Stroke/transient ischemic attack/thromboembolism	0.50	9.72	−19.65–2.89
Vascular disease	1.35	9.60	−19.71–4.51
Age 65–74 years	−0.11	1.04	−1.15–0.59
Female	0.08	0.85	−0.93–1.47

β: logistic regression coefficient; CI: confidence interval. Adjusted β were obtained with bootstrap methods based on 1000 replications.

**Table 4 medicina-56-00073-t004:** Incidence rates and incidence rate ratio of ischemic stroke for patients with rheumatoid arthritis according to the new RA-CHADSV score (N = 592).

RA-CHADSV Score	Events	Person-Years	IR (/10,000 Person-Years)	IRR (95% CI) *p* Value	Adjusted IRR * (95% CI) *p* Value
Continuous score	35	5902	59.30	2.82 (2.10–3.80) <0.001	2.49 (1.78–3.49) <0.001
Dichotomized score					
0	9	4318	27.79	1	1
≥1	26	1585	145.11	7.70 (3.61–16.44) <0.001	7.39 (3.42–15.74) <0.001

CI: confidence interval; IR: incidence rate; IRR: incidence rate ratio; RA-CHADSV: rheumatoid arthritis - congestive heart failure, hypertension, age ≥75 years, diabetes mellitus, stroke/transient ischemic attack/thromboembolism, and vascular disease. *Adjusted for dyslipidemia, chronic obstructive pulmonary disease, chronic kidney disease, and obesity.

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
