# Peer review of "Predicting the Risk of Ischemic Stroke among Patients with Rheumatoid Arthritis Using a Simplified RA-CHADSV Score Based on the CHA2DS2-VASc Score"

_medicina, 2020, doi:10.3390/medicina56020073_

Round 1
Reviewer 1 Report
The paper titled Predicting the risk of ischemic stroke among patients with rheumatoid arthritis using a simplified RA-CHADSV score based on the CHA2DS2-VASc score, presents an interesting approach to predicting the risk of ischemic stroke.
Although the paper is interesting, I have some major concerns:
Title
The title reflects the results presented here.
Abstract
The abstract is lacking informative conclusion. It should be written in more details.
Introduction
The authors should add some information about the numerical tools for the prediction of ischemic stroke as nowadays it is commonly used technique in this field, e.g.:
Computational fluid dynamics as an engineering tool for the reconstruction of hemodynamics after carotid artery stenosis operation: A case study. Medicina (Kaunas).2018 Jun 1;54(3). pii: E42. doi: 10.3390/medicina54030042.Material and Methods
Precise description of analysed group of patients should be included, e.g. number of analysed patients, age, sex.
In the results authors indicate that “Adjusted β were obtained with bootstrap methods based on 1000 replications.”, however there is no information in the description of statistical methods in material and methods. This part should be included.
Results
First paragraph including Table 1. Baseline characteristics of the study participants (N = 592) should be transfer to the Material and methods parts.
Conclusions
Conclusions should reflect the most important findings. Now they are too general.
Figures
Are correct.
Tables
Are correct.
Author Response
Reviewer 1, Comment No. 1:
Abstract: The abstract is lacking informative conclusion. It should be written in more details.
Response to Reviewer 1, Comment No. 1:
We appreciate the reviewer’s suggestion and have revised the abstract (Page 1, Line 18-19 & 30-31).
--------------------------------------------------------
Reviewer 1, Comment No. 2:
Introduction: The authors should add some information about the numerical tools for the prediction of ischemic stroke as nowadays it is commonly used technique in this field, e.g.:
Computational fluid dynamics as an engineering tool for the reconstruction of hemodynamics after carotid artery stenosis operation: A case study. Medicina (Kaunas).2018 Jun 1;54(3). pii: E42. doi: 10.3390/medicina54030042.
Response to Reviewer 1, Comment No. 2:
We have added two new references (ref. no. 13-14) (one of them suggested by the reviewer) in the revised manuscript (Page 2, Line 46-47).
--------------------------------------------------------
Reviewer 1, Comment No. 3:
Material and Methods: Precise description of analysed group of patients should be included, e.g. number of analysed patients, age, sex.
Response to Reviewer 1, Comment No. 3:
The characteristics of the patients are shown in Table 1 and in the Materials and Methods section.
--------------------------------------------------------
Reviewer 1, Comment No. 4:
In the results authors indicate that “Adjusted β were obtained with bootstrap methods based on 1000 replications.”, however there is no information in the description of statistical methods in material and methods. This part should be included.
Response to Reviewer 1, Comment No. 4:
We have added a new reference (no. 37) in the Statistical analysis section and the following text: “The model was validated internally with a bootstrap approach using 1,000 samples by resampling with replacement from the original sample with bias-corrected and accelerated confidence interval [37].”.
--------------------------------------------------------
Reviewer 1, Comment No. 5:
Results: First paragraph including Table 1. Baseline characteristics of the study participants (N = 592) should be transfer to the Material and methods parts.
Response to Reviewer 1, Comment No. 5:
We followed the suggestion from the reviewer and have moved the first paragraph of the Results section and the Table 1 to the Material and Methods section (Page 2, Line 73-79).
--------------------------------------------------------
Reviewer 1, Comment No. 6:
Conclusions: Conclusions should reflect the most important findings. Now they are too general.
Response to Reviewer 1, Comment No. 6:
We thank the reviewer’s suggestion and have revised the Conclusions section (Page 8, Line 221-225).

Reviewer 2 Report
The article presents a new tool for predicting stroke risk factors in people with RA. He also adds that these people have a higher probability of stroke than the rest of the population.
There are several points to clarify:
The objective of the study is not clear if it is to state that there is a greater risk of stroke for people with RA, or to develop a new assessment tool.
Line 71-72, The observational period was one year, in other sections indicates that the average was 10 years. explain this.
The introduction must be improved, since there are very few references, and it must be completed to clarify the selection of the sample, since it excludes patients with AF.
It is necessary to better explain the population studied, that is, the sample are people who suffered a stroke?
Justify in the sample because 80% are women, are there more women with RA?
line 119, 124 and 128, 153 in the present not in the past.
In table 3, explain values ​​less than 1, are stroke preventive factors?
in table 3, if you exclude sex, being mostly women, is it a bias?
in table 4, justify with literature the dichotomization between 0 and> or equal to 1
In general, the results section should be written more clearly and concisely, that is easier to understand.
In the discussion section, there are statements that are doubtful that they are from the study or other authors, lines 168-172
no bibliographic references can appear in the conclusions
explain better what this study provides, that is, if it is an evaluation tool and because it is better 6 variables than the current one.
Author Response
Reviewer 2, Comment No. 1:
The article presents a new tool for predicting stroke risk factors in people with RA. He also adds that these people have a higher probability of stroke than the rest of the population.
There are several points to clarify:
The objective of the study is not clear if it is to state that there is a greater risk of stroke for people with RA, or to develop a new assessment tool.
Response to Reviewer 2, Comment No. 1:
We appreciate the valuable comment from the reviewer and have revised the objective in the Abstract.
--------------------------------------------------------
Reviewer 2, Comment No. 2:
Line 71-72, The observational period was one year, in other sections indicates that the average was 10 years. explain this.
Response to Reviewer 2, Comment No. 2:
We appreciate the reviewer for the comment. We have revised the sentence to “The longest follow-up duration was up to 10 years.” (Page 2, Line 83-85).
--------------------------------------------------------
Reviewer 2, Comment No. 3:
The introduction must be improved, since there are very few references, and it must be completed to clarify the selection of the sample, since it excludes patients with AF.
Response to Reviewer 2, Comment No. 3:
We thank the comment from the reviewer and have added three new references (No. 10~12). We have also modified the last sentence of the introduction section to indicate that our sample was RA patients without AF (Page 2, Line 44-45).
--------------------------------------------------------
Reviewer 2, Comment No. 4:
It is necessary to better explain the population studied, that is, the sample are people who suffered a stroke?
Response to Reviewer 2, Comment No. 4:
According to CHA2DS2-VASc, patients who had suffered from a stroke were still in the study population.
--------------------------------------------------------
Reviewer 2, Comment No. 5:
Justify in the sample because 80% are women, are there more women with RA?
Response to Reviewer 2, Comment No. 5:
This observation reflects the epidemiology of rheumatoid arthritis. Generally, female patients with RA are 3–4 times more frequent than male counterparts.
--------------------------------------------------------
Reviewer 2, Comment No. 6:
line 119, 124 and 128, 153 in the present not in the past.
Response to Reviewer 2, Comment No. 6:
We have revised the tense from past to present in these sentences as suggested (Line 124, 128, 134, 138, and 157).
--------------------------------------------------------
Reviewer 2, Comment No. 7:
In table 3, explain values ​​less than 1, are stroke preventive factors?
Response to Reviewer 2, Comment No. 7:
The negative value (−0.11) in Table 3 means that younger patients (age 65–74 years) have a low risk of ischemic stroke. Therefore, in our model, patients with an age ≥ 75 years will be assigned 1 point in our RA-CHADSV score.
--------------------------------------------------------
Reviewer 2, Comment No. 8:
in table 3, if you exclude sex, being mostly women, is it a bias?
Response to Reviewer 2, Comment No. 8:
In Table 3, being female was not a significant factor in the multiple logistic regression model (the 95% confidence interval for female contains 0). Therefore, the exclusion of sex should not lead to a bias result.
--------------------------------------------------------
Reviewer 2, Comment No. 9:
in table 4, justify with literature the dichotomization between 0 and> or equal to 1
Response to Reviewer 2, Comment No. 9:
We have added a new reference (No. 26) to support our choice of the cut-off point (Page 5, Line 160).
--------------------------------------------------------
Reviewer 2, Comment No. 10:
In general, the results section should be written more clearly and concisely, that is easier to understand.
Response to Reviewer 2, Comment No. 10:
We have moved the first paragraph of the Results section and Table 1 to the Materials and Methods section. We have also corrected the term “standardized regression coefficients” to “logistic regression coefficients”.
--------------------------------------------------------
Reviewer 2, Comment No. 11:
In the discussion section, there are statements that are doubtful that they are from the study or other authors, lines 168-172
Response to Reviewer 2, Comment No. 11:
The reviewer indicated lines 168 to 172 of the Discussion section contains statements of unclear source. Lines 168 to 172 of original version of the manuscript are, in fact, Figure 2 and Table 4, which both are from our study.
--------------------------------------------------------
Reviewer 2, Comment No. 12:
no bibliographic references can appear in the conclusions
Response to Reviewer 2, Comment No. 12:
We have removed the references in the Conclusions section and moved them to the Introduction section.
--------------------------------------------------------
Reviewer 2, Comment No. 13:
explain better what this study provides, that is, if it is an evaluation tool and because it is better 6 variables than the current one.
Response to Reviewer 2, Comment No. 13:
We appreciate the reviewer’s suggestion. We indicated that the tools contain only six variables of equal weights, and the performance was comparable with the CHA2DS2-VASc score (Page 7, Line 212-214) and the score could be used to identify non-AF RA patients with an increased risk of ischemic stroke, and to help physicians in the management of these patients (Page 8, Line 224-226).

Round 2
Reviewer 1 Report
The authors fully answered my questions.
Author Response
Reviewer 1, Comment No. 1:
The authors fully answered my questions.
Response to Reviewer 1, Comment No. 1:
We sincerely appreciate the reviewer for taking the time and effort to review and improve our manuscript.
Reviewer 2 Report
The requested changes have been introduced.
It is necessary to remove the bibliographic references from the conclusions.
Author Response
Reviewer 2, Comment No. 1:
The requested changes have been introduced.
It is necessary to remove the bibliographic references from the conclusions.
Response to Reviewer 2, Comment No. 1:
We appreciate the reviewer’s comment. We have removed all bibliographic references from the conclusions in our previous revised manuscript.
The conclusion section is as the followings: “Findings from this retrospective cohort study showed that CHA2DS2-VASc score was able to moderately predict the risk of ischemic stroke in non-AF RA patients. Furthermore, a new RA-CHADSV score was developed in this study, which was based on six variables, including congestive heart failure, hypertension, age ≥ 75 years, diabetes mellitus, stroke/transient ischemic attack/thromboembolism, and vascular disease. This simple tool could be used to identify non-AF RA patients with an increased risk of ischemic stroke, and to help physicians in the management of these patients.”